# Coupled multiferroic domain switching in the canted conical spin spiral system $Mn_2GeO_4$

T. Honda[1,2,*], J.S. White[3,*], A.B. Harris[4], L.C. Chapon[5], A. Fennell[3], B. Roessli[3], O. Zaharko[3], Y. Murakami[2], M. Kenzelmann[6] & T. Kimura[1,†]

Despite remarkable progress in developing multifunctional materials, spin-driven ferro-electrics featuring both spontaneous magnetization and electric polarization are still rare. Among such ferromagnetic ferroelectrics are conical spin spiral magnets with a simultaneous reversal of magnetization and electric polarization that is still little understood. Such materials can feature various multiferroic domains that complicates their study. Here we study the multiferroic domains in ferromagnetic ferroelectric $Mn_2GeO_4$ using neutron diffraction, and show that it features a double-Q conical magnetic structure that, apart from trivial 180° commensurate magnetic domains, can be described by ferromagnetic and ferroelectric domains only. We show unconventional magnetoelectric couplings such as the magnetic-field-driven reversal of ferroelectric polarization with no change of spin-helicity, and present a phenomenological theory that successfully explains the magnetoelectric coupling. Our measurements establish $Mn_2GeO_4$ as a conceptually simple multiferroic in which the magnetic-field-driven flop of conical spin spirals leads to the simultaneous reversal of magnetization and electric polarization.

[1] Division of Materials Physics, Graduate School of Engineering Science, Osaka University, Toyonaka, Osaka 560-8531, Japan. [2] Condensed Matter Research Center, Institute of Materials Structure Science, High Energy Accelerator Research Organization, Tsukuba 305-0801, Japan. [3] Laboratory for Neutron Scattering and Imaging (LNS), Paul Scherrer Institut (PSI), Villigen CH-5232, Switzerland. [4] Department of Physics and Astronomy, University of Pennsylvania, Philadelphia, Pennsylvania 19104, USA. [5] Institut Laue-Langevin, BP 156X, Grenoble F-38042, France. [6] Laboratory for Scientific Developments and Novel Materials (LDM), Paul Scherrer Institut (PSI), Villigen CH-5232, Switzerland. * These authors contributed equally to this work. † Present address: Department of Advanced Materials Science, University of Tokyo, Kashiwa 277-8561, Japan. Correspondence and requests for materials should be addressed to M.K. (email: michel.kenzelmann@psi.ch) or to T.K. (email: tkimura@edu.k.u-tokyo.ac.jp).

The nomenclature 'multiferroics' has been originally coined for materials showing the coexistence of two or all three ferroic orders (ferroelectric, ferromagnetic and ferroelastic)[1,2], and is expanded nowadays to comprise materials showing ferroelectric and antiferromagnetic orders[3]. Extensive studies in the past decade have led to the discoveries of various new multiferroics in which their magnetic order breaks inversion symmetry and gives rise to ferroelectricity[3–7]. Among such multiferroics, however, materials showing both spontaneous magnetization $\mathbf{M}$ and electric polarization $\mathbf{P}$ simultaneously, that is, multiferroics in the original definition, are not common. One of the few systems exhibiting both spontaneous $\mathbf{M}$ and $\mathbf{P}$ is a spiral magnet with the so-called transverse conical spin structure, which consists of a cycloidal spiral spin component and a ferromagnetic component along the spin rotation axis of the cycloid[3,6]. This is an example of a so-called multi-$\mathbf{Q}$ structure in which two distinct magnetic propagation vectors $\mathbf{Q}$ coexist. In the framework of the spin–current or inverse Dzyaloshinskii–Moriya (DM) mechanism[8], the cycloidal component induces $\mathbf{P}$ in the direction perpendicular to both the spin rotation axis and the modulation wave vector. This means that $\mathbf{P}$ in transverse conical systems develops in the direction perpendicular to $\mathbf{M}$. $CoCr_2O_4$ (refs 9,10) and some hexaferrites[11–17] are examples of such conical–spiral multiferroics in which simultaneous reversals of $\mathbf{M}$ and $\mathbf{P}$ are often observed[9,10,12,16]. However, the microscopic mechanism for the simultaneous reversal has not been fully understood to date. An important result of the present paper is to show that the phenomenological model we introduce here explains the switching mechanisms we observe in the double-$\mathbf{Q}$ conical-spiral multiferroic, $Mn_2GeO_4$.

Recently, an olivine-type compound $Mn_2GeO_4$ was reported to be a rare multiferroic below 5.5 K, where both spontaneous $\mathbf{M}$ and $\mathbf{P}$ develop in the same direction[18,19]. The crystal structure of $Mn_2GeO_4$ (Fig. 1a) is described by the orthorhombic $Pnma$ space group. The $S = 5/2$ $Mn^{2+}$ ions occupy two distinct crystallographic sites (Mn1 and Mn2) octahedrally coordinated by $O^{2-}$ anions, and form sawtooth-like chains along the $b$ axis[20]. The structure contains competing magnetic interactions, which lead to various magnetic phases and a complex magnetic order in the ground state multiferroic phase[18,19,21]. A previous neutron scattering study[13] proposed a multi-$\mathbf{Q}$ magnetic order with both commensurate (C) and incommensurate (IC) components for the multiferroic state. The C component characterized by the propagation vector of (0 0 0) is described by a combination of two irreducible representations (irreps) $\Gamma^1$ and $\Gamma^3$ and allows finite $\mathbf{M}$ along the $c$ axis ($M_c$). The IC component forms a spin spiral with the propagation vector $\mathbf{Q}_{ic} = (q_h\ q_k\ 0)$, where $q_h = 0.136$ and $q_k = 0.211$. This structure is described by two irreps $D^1$ and $D^2$, and contains a cycloidal component which can be responsible for $\mathbf{P}$ along the $c$ axis ($P_c$). To the best of our knowledge, $Mn_2GeO_4$ is the only multiferroic with the $\mathbf{M}\|\mathbf{P}$ configuration where $\mathbf{P}$ can be understood as generated by a spiral spin order. Phenomenological symmetry analysis on the respective C and IC components well explained not only the simultaneous appearance of finite $|\mathbf{M}|$ and $|\mathbf{P}|$ but also the $\mathbf{M}\|\mathbf{P}$ configuration[18]. However, no clear accessible explanation for the $\mathbf{M}$–$\mathbf{P}$ coupling has been provided to date. In addition, as shown in Fig. 1b, a reversal of $\mathbf{M}$ accompanies that of $\mathbf{P}$ in the multiferroic state of $Mn_2GeO_4$, indicating a synchronized switching of both ferromagnetic and ferroelectric domains. This coupled switching mechanism still remains an unsolved issue, both microscopically and phenomenologically.

Here we report on the evolution of these domains in $Mn_2GeO_4$ by the application of electric and/or magnetic fields, and reveal one-to-one phenomenological correspondence relations between certain C and Q domains, which arise from the C and IC components in the multi-$\mathbf{Q}$ magnetic order, respectively. From this we show that the multiferroic state of $Mn_2GeO_4$ can be regarded as canted conical spin spirals, as illustrated in Fig. 1c,d. We demonstrate experimentally that the effective spiral handedness, which is usually controlled by an electric field in magnetic spiral ferroelectrics, couples not only with $\mathbf{P}$ but also with $\mathbf{M}$ via the C,Q domain switching, and its sign can be reversed by a magnetic field in $Mn_2GeO_4$. We also clarify the mechanism of both the $\mathbf{M}\|\mathbf{P}$ configuration and the simultaneous reversal of $\mathbf{M}$ and $\mathbf{P}$ observed in this unique multiferroic compound.

## Results

**Consideration of multiferroic domains.** To elucidate the coupling mechanism between the ferromagnetism and ferroelectricity in $Mn_2GeO_4$, we investigated the effects of electric and magnetic fields on domains by means of unpolarized and polarized neutron scattering measurements on single crystal samples. Both the ferromagnetic and ferroelectric domains in this compound involve several types of magnetic domains, which are ascribed to the complex multi-$\mathbf{Q}$ magnetic order with both C and IC components. As illustrated in Supplementary Fig. 1a, the C component can host two pairs of ferromagnetic domains (defined as $C_A$ domain for $\pm\Gamma^1 \oplus \pm\Gamma^3$ and $C_B$ domain for $\pm\Gamma^1 \oplus \mp\Gamma^3$). Each C domain possesses its time-reversal counterpart with the opposite sign of $M_c$ (compare upper and lower panels of Supplementary Fig. 1a). On the other hand, as illustrated in Supplementary Fig. 1b, the IC component can host two pairs of ferroelectric domains (defined as $Q_A$ and $Q_B$ domains for the propagation vectors $(q_h\ q_k\ 0)$ and $(q_h\ -q_k\ 0)$, respectively). Each Q domain possesses its space-inversion counterparts with the opposite sign of $P_c$ (or the opposite sign of spin-helicity $h$) (compare upper and lower panels of Supplementary Fig. 1b). Thus, the multiferroic state in $Mn_2GeO_4$ includes all four types of magnetic domains defined in ref. 22: orientation domains ($C_A \leftrightarrow C_B$), time-reversed domains $[C(+M_c) \leftrightarrow C(-M_c)]$, configuration (or Q) domains ($Q_A \leftrightarrow Q_B$) and so-called chirality domains $[Q(+h) \leftrightarrow Q(-h)]$.

**Phenomenological coupling theory.** First, we present a phenomenological theory that describes the possible coupling terms between the ferroelectric and magnetic order parameters in the multiferroic state of $Mn_2GeO_4$ (see Methods). To describe the couplings seven order parameters are identified: ferroelectric order $P_z$, two complex order parameters $M_Q^{D1}$ and $M_Q^{D2}$ for irreps $D^1$ and $D^2$, respectively, for each IC Q domain, $Q = Q_A$ or $Q_B$ (in total four order parameters), and two C order parameters $X_1$ and $X_3$, belonging to irreps $\Gamma^1$ and $\Gamma^3$, respectively. $M_Q^{D1}$ and $M_Q^{D2}$ represent the amplitude of the magnetization distribution which transforms according to the irreps $D^1$ and $D^2$ for the space group $Pnma$ and $\mathbf{Q}_{ic} = (q_h\ q_k\ 0)$ (ref. 18), for example, $M_Q^{D1}$ is even under the glide operation perpendicular to the $z$ axis ($m_z$), and $M_Q^{D2}$ is odd under $m_z$. $X_1$ describes 180° domains of the dominant C antiferromagnetic component while $X_3$ corresponds to the ferromagnetic component with magnetization along the $z$ axis ($M_z$). Here the $z$ axis corresponds to the $c$ axis in $Mn_2GeO_4$. Supplementary Table 1 and Supplementary Note 1 describe the symmetry operations of the space group and the transformation properties of these order parameters, respectively. We construct coupling terms that the reader can check are invariant under these transformation properties by using Supplementary Eqs. 1 − 7 in Supplementary Note 1, and this leads to three

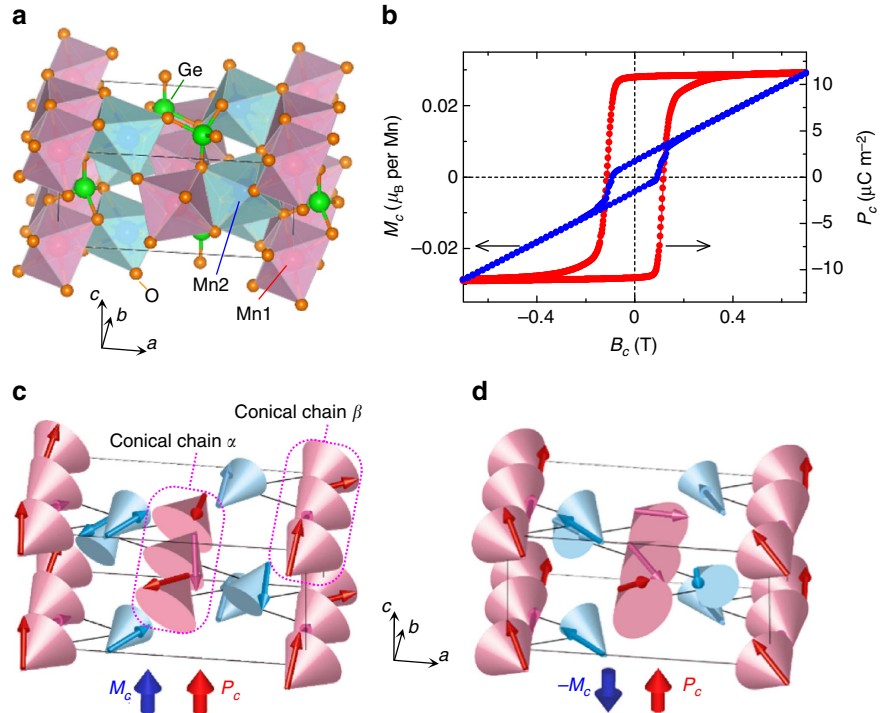

**Figure 1 | Structures and multiferroicity of Mn$_2$GeO$_4$.** (**a**) Crystal structure of Mn$_2$GeO$_4$. (**b**) Magnetization along the $c$ axis ($M_c$) and electric polarization along $c$ ($P_c$) as a function of magnetic field applied along $c$ ($B_c$) at 4.5 K for a single crystal of Mn$_2$GeO$_4$. Before the measurement of $P_c$, the specimen was cooled from high temperature into the multiferroic phase in positive electric and magnetic fields. The figure is adapted with permission from ref. 18. (**c,d**) Magnetic structures in the multiferroic state of Mn$_2$GeO$_4$ (the most probable magnetic point group: 2 with 2$_1$ axis along $c$). Two of the possible magnetic domains are illustrated: (**c**) (C$_A$, Q$_A$) domain [ $+\Gamma^1 \oplus +\Gamma^3$, ($q_h$ $q_k$ 0)] and (**d**) (C$_B$, Q$_B$) domain [ $+\Gamma^1 \oplus -\Gamma^3$, ($q_h$ $-q_k$ 0)]. Red and blue arrows on cones denote Mn1 and Mn2 moments, respectively. In the magnetic structure, two types of conical spin chains extend along the $b$ axis (conical chains $\alpha$ and $\beta$) and are placed alternately in a staggered arrangement.

such terms given by

$$U = ir' P_z \left[ \left( M_{QA}^{D1\,*} M_{QA}^{D2} - M_{QA}^{D1} M_{QA}^{D2\,*} \right) - \left( M_{QB}^{D1\,*} M_{QB}^{D2} - M_{QB}^{D1} M_{QB}^{D2\,*} \right) \right],$$

(1)

$$V = s X_1 X_3 \left( \left| M_{QA}^{D1} \right|^2 - \left| M_{QB}^{D1} \right|^2 \right) + s' X_1 X_3 \left( \left| M_{QA}^{D2} \right|^2 - \left| M_{QB}^{D2} \right|^2 \right),$$

(2)

$$W = ir X_1 X_3 P_z \left( M_{QA}^{D1\,*} M_{QA}^{D2} - M_{QA}^{D1} M_{QA}^{D2\,*} + M_{QB}^{D1\,*} M_{QB}^{D2} - M_{QB}^{D1} M_{QB}^{D2\,*} \right).$$

(3)

Here $s$, $s'$, $r$, and $r'$ are real numbers. $U$ describes the coupling between the spin-helicity and the electric polarization, and is equivalent to the trilinear coupling term developed for Ni$_3$V$_2$O$_8$ (ref. 23) and TbMnO$_3$ (ref. 24). The spin-helicity $h$ is defined as $h \propto i \left( M_Q^{D1} M_Q^{D2\,*} - M_Q^{D1\,*} M_Q^{D2} \right) / \left| M_Q^{D1} M_Q^{D2\,*} \right|$, and is of opposite sign for the Q$_A$ and Q$_B$ domains. Nevertheless, each Q domain has

the same handedness of spin cycloidal component projected onto the $bc$ plane, which induces $P_z$ with the same sign. $V$ describes the coupling between the C domains and the Q domains, and $W$ describes the coupling between the magnetization and the electric polarization. These three coupling terms ($U$, $V$ and $W$) define strict relationships between the order parameters. Table 1 lists the relationships between the signs of the respective order parameters achieved after various field cooling conditions. From these couplings, one can anticipate the effects of ferromagnetic and ferroelectric domain switching on the respective order parameters. Importantly, a reversal of $X_3$ ($= M_z$) allows not only

a concomitant reversal of $P_z$ but also a switching between the Q$_A$ and Q$_B$ domains. We will provide experimental evidence for the presence of all three coupling terms.

**Electric-field-cooling effect on spin-helicity.** To prove the existence of the $U$ coupling term (equation (1)), we studied the effect of a cooling electric field $E_{cool}$ on spin-helicity in the Q domains using polarized neutron diffraction (spherical neutron polarimetry (SNP)) at TASP, Paul Scherrer Institut (PSI). Details of the experiments are fully described in Methods. The intensity profiles of the ( $\pm 2 \mp q_h$ 1 $- q_k$ 0) peaks were measured in both the non-spin-flip ($z,x$) and spin-flip ($z, -x$) polarization channels. From the difference in the intensities observed in each polarization channel, the relative proportions of the two helicity domains within each Q domain can be determined directly (see Methods). These measurements were done in zero electric field at 2 K, after having cooled the specimen from 10 to 2 K under various $E_{cool}$ along the $c$ axis. In Fig. 2a–d we show the wave-vector dependence of the scattering from the Q$_A$ domain peak ($2 - q_h$ 1 $- q_k$ 0) after cooling under $E_{cool} = \pm 0.5$ and $\pm 1.5$ MV m$^{-1}$. The results show a strong sensitivity to both the neutron polarization state and size of $E_{cool}$, with the difference between the non-spin-flip and spin-flip scattering being larger in the data for $E_{cool} = \pm 1.5$ MV m$^{-1}$. Furthermore, by switching the sign of $E_{cool}$ the polarization-dependent magnitude of the spin-flip and non-spin-flip intensities becomes reversed. Similar data are also obtained for the Q$_B$ domain. Importantly, these results show that the $P_{zx}$ elements of the polarization matrix (see Methods) can be made finite, with both the magnitude and sign tunable by the electric-field-cooling condition. This confirms the magnetic structure in the multiferroic state of Mn$_2$GeO$_4$ to include a spin spiral

**Table 1 | Relationships between the signs of order parameters.**

| Condition before measurement | $h[Q_A, Q_B]$ | $P_z$ | $X_1$ ($\Gamma^1$) | $X_3 = M_z$ ($\Gamma^3$) | $X_1 X_3$ ($C_A$ or $C_B$) | $Q$ ($Q_A$ or $Q_B$) |
|---|---|---|---|---|---|---|
| $+E+B$ cool | $[-, +]$ | $+$ | $+(-)$ | $+$ | $+(-)$ | $Q_A(Q_B)$ |
| $-B$ sweep | $[+, -]$ | $-$ | $+(-)$ | $-$ | $-(+)$ | $Q_B(Q_A)$ |
| $+E-B$ cool | $[-, +]$ | $+$ | $-(+)$ | $-$ | $+(-)$ | $Q_A(Q_B)$ |
| $+B$ sweep | $[+, -]$ | $-$ | $-(+)$ | $+$ | $-(+)$ | $Q_B(Q_A)$ |
| $-E+B$ cool | $[+, -]$ | $-$ | $+(-)$ | $+$ | $+(-)$ | $Q_A(Q_B)$ |
| $-B$ sweep | $[-, +]$ | $+$ | $+(-)$ | $-$ | $-(+)$ | $Q_B(Q_A)$ |
| $-E-B$ cool | $[+, -]$ | $-$ | $-(+)$ | $-$ | $+(-)$ | $Q_A(Q_B)$ |
| $+B$ sweep | $[-, +]$ | $+$ | $-(+)$ | $+$ | $-(+)$ | $Q_B(Q_A)$ |

The signs are determined by considering that an electric field $E$ and a magnetic field $B$ directly act on $P_z$ and $M_z$, respectively, and by the $U$, $V$ and $W$ invariants in equations (1)–(3). In the table, square brackets show the sign of $h$ in each $Q$ domain. The sign '$+(-)$' denotes that either '$+$' or '$-$' domain is possible. However, within each experimental condition, the signs in and out of parentheses couple with those in and out of parentheses for other order parameters. Here, '$+E+B$ cool' means the field-cooled condition with positive $E$ and $B$ that are large enough to pole the system. '$B$-sweep' denotes that $B$ was swept to an opposite value once after the cooling procedure of the upper row. One-to-one correspondence between $h$ and $P_z$ is ascribed to the $U$ term (equation (1)) and that between $X_1 X_3$ and $Q$ originates from the $V$ term (equations (2)). In the column of $X_1 X_3$, '$+$' and '$-$' refer to $C_A$ and $C_B$ commensurate domains, respectively. The relationships between the respective order parameters satisfy all experimental results (Table 2 and Supplementary Table 2).

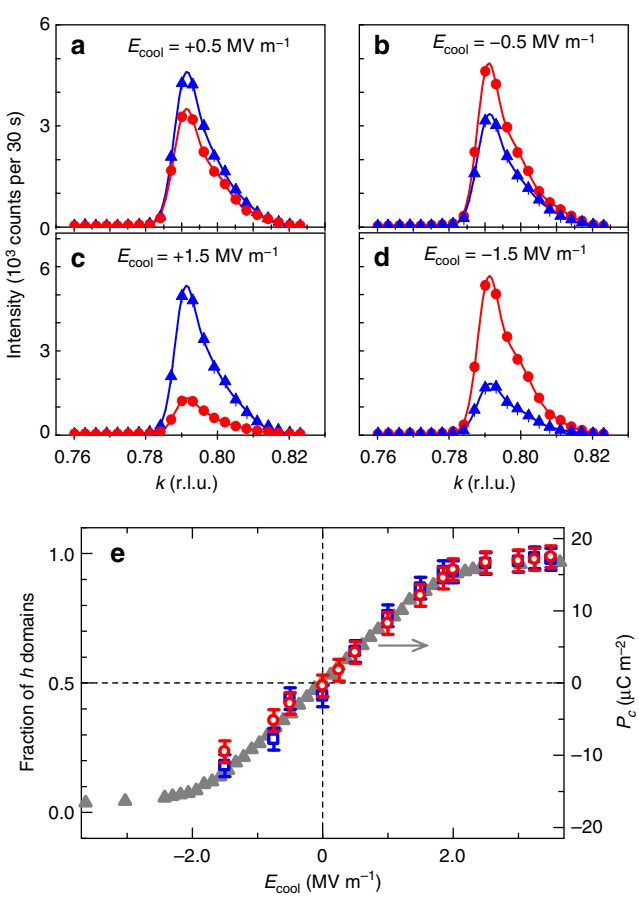

**Figure 2 | Electric-field-cooling effect on spin-helicity.** (**a**–**d**) Polarized neutron scattering as a function of wave-vector transfer along the $(0,k,0)$ wave-vector direction of the $(2 - q_h\ 1 - q_k\ 0)$ magnetic reflection in the $Q_A$ domain. The measurements were done at 2 K in the absence of electric field after cooling the specimen from 10 to 2 K at the cooling electric field $E_{cool} = +0.5$ (**a**), $-0.5$ (**b**), $+1.5$ (**c**) and $-1.5$ MV m$^{-1}$ (**d**) along the $c$ axis. Red circles and blue triangles denote the diffracted intensities of non-spin-flip $(z,x)$ and spin-flip $(z,-x)$ channels, respectively. (**e**) Fractions of the spin-helicity $h$ domains as a function of $E_{cool}$. Red open circles and blue open squares show the fraction of $-h$ domain $[p(-h)]$ in the $Q_A$ domain and that of $+h$ domain $[p(+h)]$ in the $Q_B$ domain, respectively. For comparison, the $E_{cool}$ dependence of $P_c$ is also plotted (grey triangles). Error bars in (**e**) are defined as the difference between positive and negative polarization matrices.

component with a spin-helicity that can be selected by an electric field.

We measured the spin polarization dependence of the magnetic scattering in more detail by examining the full polarization matrices for various peaks in the $Q_A$ and $Q_B$ domains (Methods). Using both the MuFIT program[25] and the magnetic structure models reported previously[18], we modelled the polarization matrix data to quantitatively determine the fractions of the spin-helicity domains ($\pm h$ domains) within each of the $Q_A$ and $Q_B$ domains, and for various $E_{cool}$. Figure 2e shows the $h$ domain fractions ($p(-h)$ and $p(+h)$ in the $Q_A$ and $Q_B$ domains, respectively), as a function of $E_{cool}$. For comparison, the $E_{cool}$ dependence of $P_c$ measured in the same condition is also plotted in Fig. 2e. The $h$ domain fraction in each $Q$ domain behaves in the same way, namely increasing in proportion with positive $E_{cool}$ up to 2 MV m$^{-1}$ and saturating above 2 MV m$^{-1}$. The $E_{cool}$ dependence of the domain fraction matches up nicely with that of $P_c$. Thus, the $Q_A$ and $Q_B$ domains coexist even after the electric-field cooling and contribute to the ferroelectric polarization, with their spin-helicity (or $h$ domain fraction) equivalently tuned by $E_{cool}$. These observations, in particular the opposite sign of $h$ in the two domains, are entirely consistent with the presence of the $U$ coupling term. It is also worth pointing out that large $E_{cool}$ ($>2$ MV m$^{-1}$) is required to obtain single spin-helicity in each $Q$ domain. In most spin spiral ferroelectrics in single crystal form, an order of magnitude smaller $E_{cool}$ is enough to pole specimens (for example, $<0.2$ MV m$^{-1}$ in Ni$_3$V$_2$O$_8$ (ref. 26)).

**Correlation between the C and Q domains.** Next, we show the results of unpolarized neutron scattering measurements done at TriCS, PSI. Note that the unpolarized neutron scattering technique cannot distinguish a pair of time-reversed 180° domains or a pair of helicity domains. From this technique, however, we obtain quantitative information about the orientation domains ($C_A \leftrightarrow C_B$) and the configuration domains ($Q_A \leftrightarrow Q_B$)[27]. In addition, by carefully examining the field-response of the respective domain populations, we also elucidate the correlation between the C and Q domains, that is, the coupling mechanism between ferromagnetism and ferroelectricity. Before the measurements, the specimen was cooled from 10 to 2 K at $E_{cool} = +3$ MV m$^{-1}$ and $B_{cool} = +1.5$ T (magnetoelectric (ME) cooling). These cooling fields are large enough to fully pole both the ferromagnetic and ferroelectric domains, which is confirmed by measurements of **M** and **P** or by checking the chiral scattering terms of the polarization matrix obtained from SNP experiments (see Figs 1b and 2e). After the ME cooling, both the electric field $E$

**Table 2 | Spin-helicity and G parameters in polarization matrix.**

| Q domain | Reflection | Cooled at $+E_{cool}$ and $+B_{cool}$ | | | After B-sweep | | |
|---|---|---|---|---|---|---|---|
| | | h | $\mathcal{P}_{yx}$ | $\mathcal{P}_{zx}$ | h | $\mathcal{P}_{yx}$ | $\mathcal{P}_{zx}$ |
| $Q_A$ | $(2 - q_h\ 1 - q_k\ 0)$ | − | − 0.66(1) | − 0.73(1) | + | 0.79(1) | 0.82(1) |
| $Q_B$ | $(-2 + q_h\ 1 - q_k\ 0)$ | + | 0.68(1) | 0.64(1) | − | − 0.82(2) | − 0.80(2) |

The spin-helicity h and the G parameters ($\mathcal{P}_{yx}$, $\mathcal{P}_{zx}$) of the $Q_A$ and $Q_B$ domains are determined from the intensities at the incommensurate magnetic reflections of $(2 - q_h\ 1 - q_k\ 0)$ and $(-2 + q_h\ 1 - q_k\ 0)$, respectively. The data of 'Cooled at $+E_{cool}$ and $+B_{cool}$' were obtained after the magnetoelectric cooling condition with $E_{cool} = +3\,MV\,m^{-1}$ and $B_{cool} = +1.2\,T$, and those of 'After B-sweep' were obtained after B was swept to $-1.2\,T$ once after the above cooling procedure. Both measurements were done at zero fields.

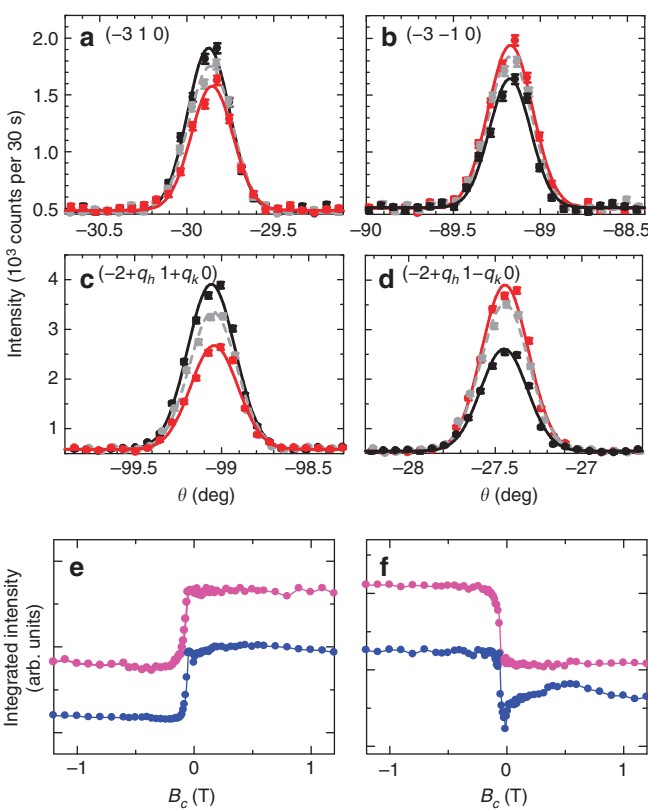

**Figure 3 | Coupled domain switching induced by a magnetic field.** Data obtained by unpolarized neutron scattering measurements. The θ-scan profiles of (**a**,**b**) the commensurate $(-3 \pm 1\ 0)$ and (**c**,**d**) the incommensurate $(-2 + q_h\ 1 \pm q_k\ 0)$ magnetic reflections. The measurements were done at 4.5 K. Black and red closed symbols denote the data measured after the ME cooling and after the reversal of **M** and **P**, respectively (see text). For comparison, the data measured after zero-field cooling are also plotted (grey symbols). Error bars are defined as the square root of counts. (**e**,**f**) Integrated intensity of the commensurate $(1 \pm 1\ 0)$ and the incommensurate $(\pm q_h\ 1 + q_k\ 0)$ magnetic reflections as a function of magnetic field along the c axis, $B_c$. Pink and blue marks in (**e**) denote the data of the $(1 - 1\ 0)$ and the $(q_h\ 1 + q_k\ 0)$ magnetic reflections, respectively. Pink and blue marks in (**f**) represent the data of the $(1\ 1\ 0)$ and the $(-q_h\ 1 + q_k\ 0)$ magnetic reflections, respectively. These data were taken while sweeping a magnetic field from positive to negative $B_c$.

and magnetic field B were swept to zero at 2 K, and then we measured integrated intensities of 50 C and 70 IC magnetic peaks. For instance, θ-scan profiles of the $(-3\ 1\ 0)$, $(-3\ -1\ 0)$, $(-2 + q_h\ -1 + q_k\ 0)$ and $(-2 + q_h\ 1 - q_k\ 0)$ reflections are shown in Fig. 3a–d, respectively (black symbols). Subsequently, B was swept to $-1.5\,T$ and then set to zero at 4.5 K. This B-sweeping procedure causes the reversal of **M** and **P**. Then the same measurements were done and the integrated intensity data of the same magnetic peaks were obtained again. Red symbols in

Fig. 3a–d show the θ-scan profiles of the above-mentioned reflections after the reversal of **M** and **P**. By comparing the data before and after the reversal of **M** and **P**, it is apparent that the intensities of the $(-3\ 1\ 0)$ and $(-2 + q_h\ -1 + q_k\ 0)$ reflections are suppressed after the reversal while those of the $(-3\ -1\ 0)$ and $(-2 + q_h\ 1 - q_k\ 0)$ reflections are enhanced. In fact, as shown in Fig. 3e,f, a stepwise change in the integrated intensity of both the C $(1 \pm 1\ 0)$ and the IC $(\pm q_h\ 1 + q_k\ 0)$ magnetic reflections is observed at the magnetic field where the reversal of **M** and **P** occurs.

As described in Methods, refinement of the integrated intensity data and determination of the respective domain fractions were done using both the FullProf program[28] and the models proposed in ref. 18. From the analysis, the domain fractions were estimated to be $p(C_A) : p(C_B) = 0.60(1) : 0.40(1)$ and $p(Q_A) : p(Q_B) = 0.600(3) : 0.400(3)$ before the reversal while those after the reversal were $p(C_A) : p(C_B) = 0.33(1) : 0.67(1)$ and $p(Q_A) : p(Q_B) = 0.371(2) : 0.629(2)$. Thus, the domain fractions of the $C_A$ and $C_B$ domains are nearly the same with those of the $Q_A$ and $Q_B$ domains, respectively. This result suggests that the C and IC domains are strongly entangled, and that the $C_A$ ($C_B$) domain is always coupled with the $Q_A$ ($Q_B$) domain. Furthermore, the domain fraction relationships between the domains A and B are reversed in both the C and Q domains after the reversal of **M** and **P**. These coupling features were observed in all the data for various field cooling conditions, as summarized in Supplementary Table 2. This indicates that a domain switching occurs not only between the $C_A$ and $C_B$ domains but also between $Q_A$ and $Q_B$ domains when the simultaneous reversal of **M** and **P** takes place. For a given sign of $X_1$, $C_A$ and $C_B$ domains are distinguished by the sign of $X_3$, and the presence of coupling term V (equation (2)) dictates a coupling of the Q domains with the C domains– consistent with our observations.

**Magnetic-field effect on spin-helicity and domain switching.** We now show the results obtained by polarized neutron diffraction (PND) measurements at D3, ILL (see Methods). As listed in Table 2, the signs of the G polarization matrix parameters ($= \mathcal{P}_{yx}$, $\mathcal{P}_{zx}$) for magnetic peaks in the $Q_A$ and $Q_B$ domains are opposite to each other after the same ME cooling procedure, and are reversed after B-sweeping, that is, the **M** reversal accompanied by the **P** reversal. This switching of the average spin-helicity with magnetic field arises due to a switching of the Q domain populations and not due to the switching of the helicity within a Q domain: it is this change of the Q domains that leads to the observed change of the spin-helicity because the $Q_A$ and $Q_B$ domains have opposite helicity according to coupling term U. This leaves the helicity-dependent term in the coupling term W overall invariant, (equation (3)), so that W describes the coupling between ferromagnetic and ferroelectric orders in $Mn_2GeO_4$.

**Discussion**

Based on these experimental results, we conclude that the full magnetic structure in the multiferroic state of $Mn_2GeO_4$ is of the

double-$Q$ type, being namely a superposition of the $C_A$ and $Q_A$ domains (or the $C_B$ and $Q_B$ domains). This means that only the following eight-types of C, Q domain combinations are possible: $[C_A(\pm M_c), Q_A(\pm P_c)]$, $[C_A(\pm M_c), Q_A(\mp P_c)]$, $[C_B(\pm M_c), Q_B(\pm P_c)]$, and $[C_B(\pm M_c), Q_B(\mp P_c)]$ (the double sign applies in the same order as written). The sign of $X_1$ determines whether **M** is parallel or antiparallel to **P**. In Fig. 1c,d we show two of the eight possible double-**Q** domains. The resulting magnetic structures can be simplified by considering them to include two types of conical spin chain ($\alpha$ and $\beta$) formed by Mn1 moments. The simplified structure is illustrated in Fig. 4a,b. In each conical chain, the Mn moments form a cone with the cone axis (∥ vector sum of the Mn moments in each chain) making a finite angle with the $a$, $b$, and $c$ axes. The coupling between the conical chains $\alpha$ and $\beta$ is canted antiferromagnetic. Though the net moments along the $a$ and $b$ axes are cancelled out, the $c$ axis component remains finite. As a result, a net magnetization $\mathbf{M}_{net}$ develops along the $c$ axis, as schematically illustrated in Fig. 4b. Furthermore, each conical chain comprises a cycloidal

component which allows a local **P** in the direction perpendicular to both the component of the propagation vector ∥$b$, and the spin rotation axis through the inverse DM mechanism[8]. The vector sum of the local **P** in the conical chains $\alpha$ and $\beta$ is aligned along the $c$ axis, and a net electric polarization $\mathbf{P}_{net}$ develops along the $c$ axis (Fig. 4b). Thus, the multiferroic state of $Mn_2GeO_4$ can be regarded simply as 'canted conical spin spirals' in which magnetically-induced electric polarization develops in the direction parallel to magnetization.

Now, we turn to the mechanism of the simultaneous reversal of magnetization and electric polarization by applying $B$ along $c$ ($B_c$) whose sign is opposite to that of $M_c$. As mentioned above, the results of unpolarized neutron scattering experiments suggest that domain switching occurs between the $(C_A, Q_A)$ and $(C_B, Q_B)$ domains when the reversal of **M** and **P** takes place. Schematics of the change in a cone axis upon the reversal of **M** and **P** (or domain switching) are illustrated in Fig. 4c. Starting from the $(C_A(+M_c), Q_A(+P_c))$ domain with positive $M_c$ and $P_c$ (Top left panel of Fig. 4c), a magnetic field antiparallel to $M_c$ will lead to the domain switching into the $(C_B(-M_c), Q_B(-P_c))$ domains with negative $M_c$ and $P_c$ (Top right panel of Fig. 4c). As displayed in Fig. 4c, each of the eight possible domains can be transformed into only one of the other domains by applying $B_c$. Such a $B$-induced domain switching can be viewed as a flop of the cone axis. Note that the cone axis flops across not only the $ab$ plane but also the $bc$ plane during the switching. The cone-axis flop across the $ab$ plane leads to the reversal of **M** along the $c$ axis while the cone-axis flop across the $bc$ plane gives rise to the reversal of **P** along the $c$ axis. This is because the effective spiral handedness of the cycloidal component in the cone projected onto the $bc$ plane reverses after the flop. The reversal of the handedness of the spin cycloid component with propagation vector along the $b$ axis and the rotation axis along the $a$ axis causes the reversal of magnetically-induced **P** along the $c$ axis in the framework of the inverse DM mechanism[8]. This scheme is also consistent with no sign change in the spin-helicity $h$ after the reversal of **M** and **P**, which was revealed by our unpolarized and polarized neutron scattering measurements. Thus, the simultaneous reversal of **M** and **P** observed in $Mn_2GeO_4$ is well explained in the light of a flop of the cone axis in the conical spin chains.

Phenomenologically, the **M**-**P** coupling can be described by the coupling term $W$ (equation (3)), as we will show now: from the coupling term $V$ (equation (2)) we know that a reversal of $X_3$ (or the magnetization) leads to switching of the Q domains (from $Q_A$ to $Q_B$ or vice versa). The coupling term $W$ dictates that the two different Q domains couple with the same helicity to both magnetization and ferroelectricity. Experimentally, we established that a reversal of the magnetization leads to a conservation of the spin-helicity and to a switch of the Q domains, which leaves the term in $W$ describing the magnetic structure invariant under a reversal of the magnetization. The term $W$ describes thus a direct coupling between the ferroelectric polarization and the magnetization—consistent with the microscopic scenario explained above.

In summary, unpolarized and polarized neutron scattering experiments have been carried out on single crystals of the olivine-type $Mn_2GeO_4$ to elucidate a unique coupling mechanism between ferromagnetism and ferroelectricity observed in this multiferroic compound. We investigated the double-**Q** magnetic structure of the multiferroic state by selectively preparing and controlling the populations of various types of magnetic domains. We have shown that the combination of canted commensurate and spin spiral incommensurate magnetic components offer unconventional magnetoelectric couplings such as the magnetic-field control of Q domain switching and electric polarization

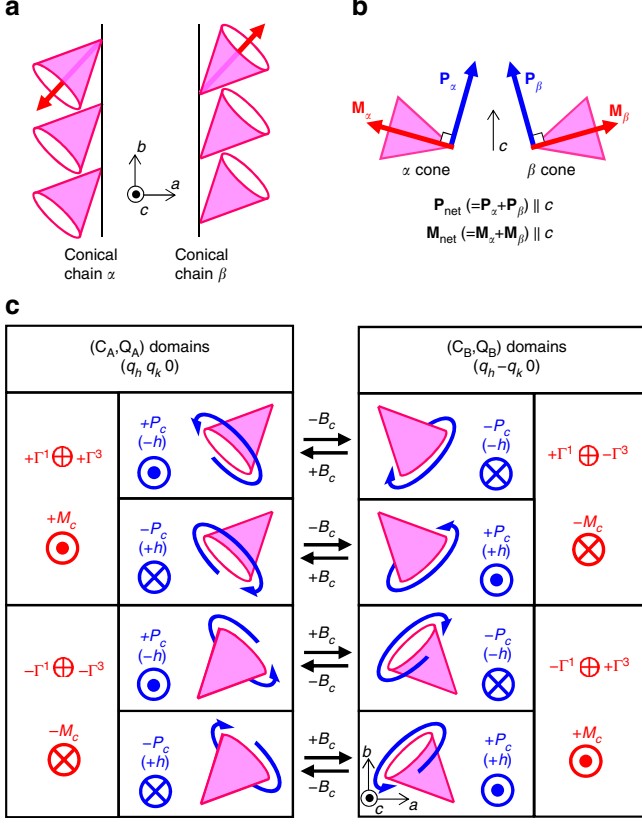

**Figure 4 | Schematics of coupled multiferroic domain switching.**
(**a**) Simplified conical magnetic structure viewed along the $c$ axis. Red arrows denote conical axes. The structure consists of two-types of conical spin chains extended along the $b$ axis (conical chains $\alpha$ and $\beta$). (**b**) The schematic configurations of local magnetization $\mathbf{M}_{\alpha,\beta}$ (red arrows) and magnetically-induced electric polarization $\mathbf{P}_{\alpha,\beta}$ (blue arrows) for each conical chain are illustrated. In each conical chain, $\mathbf{M}_{\alpha,\beta}$ and $\mathbf{P}_{\alpha,\beta}$ are aligned in directions parallel and perpendicular to the cone axis, respectively. The coupling between the two types of chains is canted antiferromagnetic, which results in both net magnetization $\mathbf{M}_{net}$ and electric polarization $\mathbf{P}_{net}$ along the $c$ axis. (**c**) Schematic illustrations of coupled multiferroic domain switching by a magnetic field along the $c$ axis, $B_c$. Blue circular arrows represent the sense of spin spiral, which corresponds to the sign of the spin-helicity $h$. The application of $B_c$ causes a flop of the cone axis, which leads to coupled multiferroic domain switching.

reversal without a change of spin-helicity. The double-$Q$ magnetic structure of multiferroic $Mn_2GeO_4$ can be regarded as canted conical spin spirals, which well explains the parallel configuration of spontaneous magnetization and electric polarization. Furthermore, we revealed that the simultaneous reversal of magnetization and electric polarization observed in this multiferroic can be understood in terms of a flop of the cone axis in the conical spin chains. The present study describes how magnetization and ferroelectric polarization in spin-driven ferroelectrics with conical magnetic order can be coupled on a phenomenological level in the case where they are parallel to each other. Furthermore, it stands out as a text-book case of how complex and seemingly unexpected coupling terms can be engineered in such spin-driven ferroelectrics. The essence of the present phenomenological discussion can be also applied to other conical-spin-driven ferroelectrics.

## Methods

**Sample preparation and measurement of electric polarization.** Single crystals of $Mn_2GeO_4$ were grown by the floating zone technique, as reported previously[14]. For measurements of neutron diffraction and electric polarization **P**, the crystals were cut into thin plate-shaped specimens. The widest faces of these specimens were perpendicular to the $c$ axis in order to apply an electric field along the ferroelectric $c$ axis. For the measurements of **P**, a plate-shaped specimen with the dimension approximately $1\,mm^2 \times 45\,\mu m$ was used, and silver electrodes were vacuum-deposited on the surfaces of the specimen. To obtain **P** as a function of cooling electric field, pyroelectric current was measured with an electrometer after cooling the specimen from high temperature into the multiferroic phase in various cooling electric fields applied along the $c$ axis.

**Phenomenological coupling theory.** To develop the phenomenological coupling theory, we first determined the symmetry properties of the magnetic and ferroelectric order parameters found experimentally. The symmetry properties of the magnetic order parameters are determined by their irreducible representation, and lead directly to the transformation properties of the order parameters given in Supplementary Eqs $2-7$ of Supplementary Note 1. The transformation property of the ferroelectric polarization is given in Supplementary Eq. 1 of Supplementary Note 1, similar to the procedure used in ref. 29. Using these transformation properties, we construct coupling terms that are invariant under the space group symmetry elements, and those relevant to $Mn_2GeO_4$ are listed in equations (1)–(3). A full account of the phenomenological coupling theory is described in ref. 30.

**Polarized neutron diffraction.** Polarized neutron diffraction (PND) measurements were carried out using both the cold neutron triple-axis spectrometer TASP ($\lambda = 3.189\,Å$) equipped with the MuPAD device[31] at the PSI, Switzerland, and the hot diffractometer D3 ($\lambda = 0.825\,Å$) equipped with the CryoPAD device[32] at the Institut Laue-Langevin, France. All experiments were carried out on single crystal samples of approximate masses 50–100 mg, and cut into thin plates of approximate cross-section $10 \times 20 \times 1\,mm^3$, with thin dimension parallel to the ferroelectric $c$ axis. The samples were always mounted with a $[100] - [010]$ horizontal scattering plane, which allowed access for SNP measurements over a range of commensurate (C) and incommensurate (IC) magnetic peak positions.

**PND measurements at TASP.** For the TASP experiment, a 10 nm layer of Cr and 50 nm layer of Au were sequentially evaporated onto the surfaces of the plate-like sample for making electrical connections. The sample was then oriented and mounted on an aluminium spacer positioned at the end of a bespoke sample stick designed for the application of high voltages (up to 5 kV) at cryogenic temperatures[33]. The high-voltage lines were connected directly to the Cr–Au sample electrodes using silver paint, so that electric fields could be applied along the ferroelectric $c$ axis. The sample space at the end of the stick was indium-sealed and covered by an Al vacuum can. The sample space was evacuated in order to avoid electric-field breakdown across the sample during the measurements. The high voltage stick with pumped sample space was then installed into a standard Orange cryostat (base temperature of 2 K), which itself was installed into the MuPAD device at the TASP beamline. Polarization matrix measurements were carried out at 2 K, after either zero-field cooling or electric-field cooling through the multiferroic transition at 5.5 K. When at the base temperature, the electric field could be removed for the SNP measurements. We observed no effect on the resulting data due to the electric field removal in the multiferroic phase.

**PND measurements at D3.** For the D3 experiments, a plate-like sample (without pre-deposited electrodes) was installed onto a different bespoke 5 kV high voltage sample stick. On this stick the application of electric fields along the $c$ axis is

achieved by a potential difference applied across two parallel Al plates. The sample was fixed to one plate by a silver epoxy, and the other plate positioned close to, but not touching the other sample surface. Thus the actual electric field across the sample was not exactly the same as the applied voltage. By comparing between similar electric-field cooling measurements as done at TASP we could estimate the absolute electric field scale in the D3 experiments. Similarly as for the TASP experiments, the sample space of the sample stick used at D3 was also indium-sealed by an Al can and the sample space evacuated. The pumped sample stick was then installed into a 'thin tail' Orange cryostat (base temperature of 3.5 K), which could then be installed into the CryoPAD device at the D3 beamline. SNP measurements were done after either zero-field cooling or electric-field cooling through the multiferroic transition, similarly as at TASP.

In addition at D3, an electromagnet (maximum applied field $\pm 1.2\,T$) was available. By installing the thin-tail cryostat into the electromagnet, vertical magnetic fields could be applied along the $c$ axis, in addition to electric fields. Thus at D3 we could investigate the effects of magnetoelectric cooling (simultaneous magnetic field- and electric field-cooling) into the multiferroic phase, and also magnetic field sweeping from positive to negative magnetic fields within the multiferroic phase. Since the electromagnet cannot be integrated with the CryoPAD device, each sample state preparation involving the magnetic field was done 'offline;' by using the beamline crane we could seamlessly interchange the high voltage stick and cryostat ensemble between the CryoPAD device and the electromagnet stationed nearby. Thus, once the sample had been either poled to 3.5 K through the multiferroic transition under simultaneously applied electric- and magnetic-fields, or the magnetic field was swept between $\pm 1.2\,T$ at the slightly elevated temperature of 4.5 K, all fields were then removed before the cryostat and sample stick were installed back into the CryoPAD device for the SNP measurements of the newly prepared state.

**Spherical neutron polarimetry technique.** Both of the MuPAD and CryoPAD devices allowed for SNP measurements of magnetic Bragg scattering at TASP and D3, respectively. SNP is a technique that, given an incident neutron beam polarized along an arbitrary direction, enables the full measurement of the direction of neutron spin polarization after the scattering process. The results are expressed as a polarization matrix

$$\mathcal{P}_{ij} = \frac{|\langle i|\mathbf{M}_\perp(\mathbf{q})\cdot\boldsymbol{\sigma}|j\rangle|^2 - \left|\langle i|\mathbf{M}_\perp(\mathbf{q})\cdot\boldsymbol{\sigma}|\bar{j}\rangle\right|^2}{|\langle i|\mathbf{M}_\perp(\mathbf{q})\cdot\boldsymbol{\sigma}|j\rangle|^2 + \left|\langle i|\mathbf{M}_\perp(\mathbf{q})\cdot\boldsymbol{\sigma}|\bar{j}\rangle\right|^2}, \tag{4}$$

where the ket $\langle i|$ describes the initial neutron spin state, the bra $|j\rangle$ describes the final neutron spin state (or $|\bar{j}\rangle$ the final spin-flip state), $\boldsymbol{\sigma}$ is the Pauli operator for the neutron spin, and $\mathbf{M}_\perp(\mathbf{q})$ is the component of $\mathbf{M}(\mathbf{q})$ that is perpendicular to the neutron scattering vector $\mathbf{q}$. Here $\mathbf{M}(\mathbf{q})$ is the magnetic structure factor. Full details of the calculation of $\mathbf{M}(\mathbf{q})$ for the different magnetic structures in $Mn_2GeO_4$, are given in the Supplement of ref. 18. The quantity $I_{ij} = |\langle i|\mathbf{M}_\perp(\mathbf{q})\cdot\boldsymbol{\sigma}|j\rangle|^2$ is the intensity of the magnetic Bragg peak observed for the incoming $i$ and outgoing $j$ spin polarizations.

The SNP technique is a particularly powerful probe of complex magnetic structures and magnetic domain distributions. For the IC spiral structure in the multiferroic phase of $Mn_2GeO_4$, and for a perfectly efficient polarized neutron scattering setup, a general polarization matrix can be written for a magnetic Bragg peak probed at $\mathbf{q}$:

$$\mathcal{P} = \begin{pmatrix} -1 & 0 & 0 \\ G & -F & 0 \\ G & 0 & F \end{pmatrix}. \tag{5}$$

Here $F = \frac{|M_z|^2 - |M_y|^2}{|\mathbf{M}_\perp|^2}$, and $G = \frac{2\Im\{M_z^*M_y\}}{|\mathbf{M}_\perp|^2}$. In this work we use the standard coordinate system where $x$ is defined as always parallel to $\mathbf{q}$, $z$ perpendicular to the scattering plane, and $y$ orthogonal to both $x$ and $z$. Consequently at each $\mathbf{q}$, $\mathbf{M}_\perp = (0, M_y, M_z)$. The $F$ and $G$ parameters of $\mathcal{P}$ are measured experimentally and compared with the values expected according to a model for magnetic structure and domain populations. The terms $G$ are the so-called 'chiral' scattering elements of $\mathcal{P}$, and by definition can only be finite when the magnetic structure has multiple modes that are out-of-phase with one another, such as is the case for non-collinear structures like cycloids, helices, and the IC spiral order in $Mn_2GeO_4$. The measured value of $G$ not only depends on the magnetic structure in the sample, but it is also sensitive to the populations of the possible domains of the magnetic structure. In the multiferroic phase of $Mn_2GeO_4$, the spin spiral magnetic structure of each Q domain has a helicity degree-of-freedom ($\pm h$) the sign of which is intimately related to the direction of ferroelectric polarization along either $+c$ or $-c$. After zero-field cooling, equal populations of opposite helicity domains are realized, which will give either $G < 0$ or $G > 0$. Thus for equal helicity domain populations the so-called 'chiral' scattering averages out and the measured value of $G = 0$, as would be observed for a collinear structure. On the other hand, as we demonstrate in our experiments at TASP, by applying an electric field along the ferroelectric $c$ axis while cooling, the helicity domain population within each Q domain can be controlled, and finite values of $G$ are observed. Thus, not only does measuring a

finite value of $G$ allow the determination of the relative fraction of spiral domains of opposite handedness (which generate ferroelectricity along either $+c$ or $-c$), but it further confirms the IC phase of $Mn_2GeO_4$ to be symmetry-breaking complex magnetic order with out-of-phase modes.

The full calculation of the expected $F$ and $G$ parameters further depends on the exact orientation of the magnetic scattering vector **q** with respect to the real-space spin distribution. If the measured **q** is exactly perpendicular to the spin rotation plane of the non-collinear spiral structure, then one obtains $G = \pm 1$ for a mono-handedness spiral state and $F = 0$. Typically however, most of our polarization matrix measurements were done at IC **q** vectors which lie at angles away from perpendicular to the spin spiral rotation plane. Consequently, even for a Q domain containing mono-handed spiral order, the magnitude of $G$ becomes $< 1$, and the magnitude of $F$ larger than 0. Nevertheless, when armed with a full model for the magnetic structure, the measured values for $F$ and $G$ at any **q** allow for a full determination of the spiral-handedness domain populations. We used the MuFIT program[25] for both analysing and fitting the SNP data to determine the magnetic domain populations.

We also briefly mention that in our SNP experiments we measured some polarization matrices of Bragg peaks due to the C magnetic orders in the various phases of $Mn_2GeO_4$. As expected for these simple **q** = 0 antiferromagnetically modulated structures, the measured matrices only had finite diagonal elements, and the $G$ parameters were always measured to be 0 within uncertainty.

**Unpolarized neutron diffraction.** Unpolarized neutron diffraction measurements were carried out using the TriCS diffractometer at PSI ($\lambda = 1.178$ Å). Similarly as for the SNP experiments described above at TASP, a plate-like $Mn_2GeO_4$ sample was mounted on the same 5 kV high voltage stick[33] so that electric fields could be applied parallel to the crystal $c$ axis. The horizontal scattering plane at TriCS contained the $[100] - [010]$ directions. The high voltage stick was installed into the variable temperature insert (VTI) of a 6 T vertical field cryomagnet (base temperature 2 K), so that both electric and magnetic fields could be applied simultaneously along the crystal $c$ axis. Unlike at D3, the cryomagnet apparatus could be installed on the beamline, and the electric- and/or magnetic-field cooling from high temperature into the multiferroic phase done 'online'. After field-cooling the sample through the multiferroic transition, all fields were then removed before data collection at 2 K. In the case of magnetic-field sweeping an already poled sample, the magnetic field sweeps were done after carefully elevating the temperature to 4.5 K. We found that at this slightly higher temperature the effect of the magnetic field sweeping on the magnetic domain populations was more pronounced than when sweeping was done at 2 K. After the sweeps the field was removed and the temperature lowered back to 2 K for the data collection.

After each multiferroic state preparation, either by electric- and/or magnetic-field cooling, or by magnetic field sweeping, we studied both the C and IC magnetic structure components that exist in this state. Typical data sets constituted integrated intensity measurements for 50 C magnetic peaks and 70 IC magnetic peaks. Refinements of the integrated intensity data and magnetic domain populations were done using the FullProf program[28] and the models proposed in ref. 18. From these refinements we determined the relative populations of the $C_A$ and $C_B$ C domain types, and the $Q_A$ and $Q_B$ IC domain types for each prepared state (with these domain types introduced in the main text), and consequently the relationships between them based on the response of their relative populations to the applied electric and magnetic fields.

**Data availability.** The data supporting the findings of this study are available from the corresponding authors on request.

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

## Acknowledgements

We thank M. Fiebig for valuable discussions. We acknowledge financial support from Grants-in-Aid for Scientific Research (No. 24244058), Grant-in-Aid for the Japan Society for the Promotion of Science Fellows (No. 24●1989), the Swiss National Centre of Competence in Research program MaNEP and the SNF under Grant Nos. 200021_126687, 200021_138018, and 20021_153451, Switzerland. This work is based on experiments performed at SINQ, PSI, Villigen, Switzerland and ILL, Grenoble, France. The work was also supported by Condensed Matter Research Center (CMRC) in KEK, Japan.

## Author contributions

T.H., J.S.W., M.K. and T.K. designed this work. T.H. and T.K. prepared single crystals of Mn₂GeO₄. T.H., J.S.W. and M.K. carried out the neutron scattering experiments with L.C.C. (D3, ILL), A.F. and B.R. (MuPAD + TASP, PSI) and O.Z. (TriCS, PSI). A.B.H. developed the phenomenological coupling theory. T.H., J.S.W., M.K. and T.K. co-wrote the paper. All authors reviewed the manuscript and agree with the results and conclusions.

**Additional information**

**Competing interests:** The authors declare no competing financial interests.

