## [Peer Review File · Nature Communications]

Reviewers' comments:

Reviewer #1 (Remarks to the Author):

In the present work, the authors focused on a rare multiferroic system--Mn₂GeO₄, whose electric polarization and magnetization are along the same direction. The authors not only provided systematic analysis of various domains as well as their evolution according to electric/magnetic field, but also developed a phenomenological theory explaining the mechanism for the M-P coupling. In such phenomenological model, the coupling between spin-helicity and P, C and Q domains, as well as M and P can be described by U, V and W terms, respectively. They further showed experimental proof for these coupling terms. In my opinion, the authors proposed a novel concept, which will be of interest to the multiferroic community. As a result, I would like to recommend the present work for publication in Nature Communications, if the authors can address my following comments:

(1) In the abstract, the authors indicated that "Despite remarkable progress towards novel multifunctional materials, single phase multiferroics featuring both spontaneous magnetization M and electric polarization P are still rare." I agree that the type-II multiferroics (with P//M) are rare. However, in my mind, the single phase multiferroics, with both P and M, are not rare, examples include BiFeO₃, PbNiO₃, hexagonal LuMnO₃&LuFeO₃, orthorhombic GdFeO₃&TbMnO₃, Strained EuTiO₃, RMn₂O₅, CuO, etc (see Dong et al, Multiferroic materials and magnetoelectric physics: symmetry, entanglement, excitation, and topology, Advances in Physics, 2015, Vol. 64, Nos. 5-6, 519-626). In these examples, the materials present both (weak) polarization and (weak) magnetization.

(2) The detailed description about the $M_{\langle Q \rangle}^{D1}$ and $M_{\langle Q \rangle}^{D2}$ ($Q=Q_A$ or Q_B) order parameters should be provided. How to get the transformations (in supplementary materials) of these four parameters also need to be explained in detail. This will be helpful to various reader to understand their phenomenological model. By the way, will it be possible to show a schematic illustration about such transformations?

(3) In Fig. 1c, I suggest the authors to mark the directions of polarization and magnetization, respectively. If possible, the (magnetic) space group or point group resulted by the corresponding magnetic configurations can also be added in Fig. 1c.

(4) Will it be possible to provide a schematic illustration to show how the inversion center is broken by the magnetic configurations in Fig 1c?

(5) In line 190, typo--"profiles of the $(-3 \ 1 \ 0)$, $(-3 \ 1 \ 0)$, $(-190 \ 2+qh \ -1+qk \ 0)$, and $(-2+qh \ 1-qk \ 0)$ reflections"

(6) Minor suggestions:

(i) In Supplementary Table 1, I understand that considering the mirror symmetry operations (m_x , m_y and m_z) are enough to derive the transformations of Pnma symmetry, because the other operations I , 2_x , 2_y and 2_z can be generated by (m_x , m_y and m_z). I still suggest the authors to indicate why considering only (m_x , m_y and m_z) symmetry operations will be enough to derive the invariant terms;

(ii) In Supplementary Table 1, I understand that the polar vectors (e.g. P) and axial vectors (e.g. M) transform in the same way under rotations, but in different ways under mirrors. I still suggest the authors to write a little bit about such transformation rule.

Please note that address the (i) and (ii) will also allow the non-expert readers to understand and check their invariant derivation without confusion.

Reviewer #2 (Remarks to the Author):

The present study of polarized & unpolarized ND in combination of a phenomenological theory on the olivine compound Mn₂GeO₄ extends the previously discoveries of M//P below T_{N3} and their reversal with magnetic field (PRL2012, JPSJ2012), and the authors address how the previous findings can be explained in a more complete way. By its own nature, it gives an apparent impression that this work is certainly a kind of incremental progresses over the initial discovery.

On the other hand, from the extensive ND study under E and B applications, to fully understand the microscopic coupling between 8 different domains is experimentally and theoretically challenging, which the authors made significant progresses in my view. It reveals the key role of the canted conical spin order and its flop with magnetic field to have both reversal of M and P with magnetic field. As the manuscript is well written (although the readers might have to carefully follow all the previous publications to fully understand due to the complicated nature of the subjects and rather concise writings of the author's), I believe the present work might be publishable in nature comm.

I have to still mention a problem in the author's view reflected in the manuscript though. As the MF research gets more and more ample, we all know that the understanding of the mechanism in each rare case of ferromagnetic ferroelectric MF becomes increasingly important. The authors seems to address well this issue in the abstract and introduction while they explain the cases of the hexaferrites and CoCr₂O₄. However, the authors seem to describe only the author's works without unbalanced citations of the other progresses in the hexaferrites, particularly the ones with Al doped compounds. I hope this can be remedied in their manuscript before being published.

Reviewer #3 (Remarks to the Author):

The article by T. Honda et al. entitled "Coupled multiferroic domain switching in the canted conical spin spiral system Mn₂GeO₄" describes a combination of high-quality polarized neutron diffraction work and a phenomenological theory on Mn₂GeO₄ that explains how the application of a magnetic field allows to simultaneously switch magnetic and ferroelectric domains. In particular, the presented data and corresponding analysis is of very high quality. As such, I recommend that the manuscript is published in Nature Communications. However, I have several comments that need to be addressed before the manuscript may be published. They are described below:

1. Section Phenomenological coupling theory: The authors write "We construct coupling terms that the reader can check are invariant under these transformation properties, and this leads to three such terms given by...". While I have no doubt that the terms are indeed invariant under these transformation properties, I don't think it is the reader's responsibility to verify this. I strongly suggest that the authors include a section in the Supplementary Material that demonstrates that this is the case.

2. Section Electric-field-cooling effect on spin-helicity: To the expert reader, familiar with the SNP technique, it is immediately clear that measuring the (z, x) and (z, -x) polarization channels provides a direct way of determining the spin helicity. However, to the broader audience, and even to most researchers familiar with neutron scattering this is not clear. After mentioning that these polarized channels were determined for varying the strength of the electrical field applied during cooling, the authors go on to describe further technical details, and only on the next page, explain that measuring these cross-sections, and eventually the full polarization matrix allows to determine the population of the two spin-helicity domains. To highlight the importance of this measurement to the non-expert reader and to improve the flow of the text, I recommend adding a statement at the beginning of the paragraph that explains that measuring the (z, x) and (z, -x) polarization channels is a direct way of determining spin helicity and is, thus, the method of choice to understand the nature of the U coupling term. I note that, for example, in the section Correlation between the C and Q domains the authors have done a much better job of explaining at the beginning of the section what kind of information can be obtained via the complimentary unpolarized neutron diffraction measurements.

3. In Fig. 2a, as well as the corresponding discussion in the text, the polarized scans of the magnetic Bragg peaks are called "k-scan profiles". I assume that the authors want to imply that they are scanning the over the peaks keeping the h and l components for the momentum transfer constant while varying k component. I think that for someone experienced in neutron diffraction

this is clear, however, for a non-expert “k-scan profiles” is a confusing name, notably, because they will not be familiar with the fact that in scattering techniques the three components of the momentum transfer in reciprocal lattice units are typically called (h, k, l). I would very much prefer to call the scans Q-scans (because in the method section the momentum transfer is called Q) or even better momentum transfer scans along the k direction. I would further replace the label in the upper left corner of Fig. 2a to be (2-qh k 0).

4. The authors highlight twice in the manuscript that no clear microscopic explanation for the simultaneous reversal of the magnetization and polarization observed in Mn₂GeO₄ and several other multiferroic materials exists to date. The discussion is written in a way that suggests that the authors established a microscopic theory that explains such switching for Mn₂GeO₄. In contrast, the theory presented in the paper is only phenomenological and based on symmetry considerations. The “microscopic” explanation provided in the discussion of the paper is based on the magnetic structure, which consists of two coupled canted antiferromagnetic conical spin-chains. The superposition of those two chains leads to net polarization and magnetization along the c axis. The neutron diffraction results described in the results part of the text show that application of a magnetic field along c leads to a flop of the cone axis in each chain, and thus simultaneous reversal of both polarization and magnetization. While the authors observe the microscopic magnetic structure of Mn₂GeO₄ and its change as function of applied magnetic and electrical field, this does not provide a microscopic explanation based on a microscopic Hamiltonian. Basically, the microscopic interactions that lead to the observed magnetic structure and its behavior in applied fields is not provided by the authors. The authors say that the polarization in the individual chains is well-explained within inverse Dzyaloshinskii-Moriya mechanism. This indeed makes sense in terms of their experimental observations. However, in that sense there is no new insights on how this simultaneous reversal of polarization and magnetization works, as this has been already proposed earlier for other materials. What is distinct about Mn₂GeO₄ seems to be rooted in details of its magnetic structure, namely that it is composed in a special superposition of two canted antiferromagnetic conical spin-chains. I thus feel that the manuscript needs some fine-tuning that is less suggestive of the authors identifying a new level of microscopic understanding of the simultaneous reversal of the magnetization and polarization in multiferroic materials.

5. For the summary paragraph, I wonder if there are some “lessons” or “guidelines” that can be extracted from the more detailed understanding of the coupling of the magnetization and electrical polarization in Mn₂GeO₄. For example, is it expected that such coupling based on coupled spin chains is found in more materials? Or is this a specific case?

Response to the reviewers' comments

We would like to sincerely thank all the reviewers for their careful reading of our manuscript as well as for constructive suggestions to improve the quality of the manuscript. In the following, we reply to the respective reviewers' comments.

Hereinafter, the comments from each reviewer appear in black italic letters while our reply is written with blue letters. Significant revisions and newly added descriptions are indicated with red letters in the manuscript.

[Response to Reviewer #1]

(1-1) *In the abstract, the authors indicated that "Despite remarkable progress towards novel multifunctional materials, single phase multiferroics featuring both spontaneous magnetization M and electric polarization P are still rare." I agree that the type-II multiferroics (with $P//M$) are rare. However, in my mind, the single phase multiferroics, with both P and M , are not rare, examples include BiFeO_3 , PbNiO_3 , hexagonal LuMnO_3 & LuFeO_3 , orthorhombic GdFeO_3 & TbMnO_3 , Strained EuTiO_3 , RMn_2O_5 , CuO , etc (see Dong et al, Multiferroic materials and magnetoelectric physics: symmetry, entanglement, excitation, and topology, *Advances in Physics*, 2015, Vol. 64, Nos. 5–6, 519–626). In these examples, the materials present both (weak) polarization and (weak) magnetization.*

Responding to the reviewer's comment, we replaced the term "single phase multiferroics" with "spin-driven ferroelectrics" in the sentence. We don't want use the term "type-II multiferroics" which is less familiar to non-expert readers. In the list raised by the reviewer includes some type-II multiferroics (or spin-driven ferroelectrics). Among them, we agree with the reviewer that some orthoferrites such as GdFeO_3 exhibits both spontaneous magnetization and polarization. However, the others such as TbMnO_3 , RMn_2O_5 , and CuO are antiferromagnets with no "spontaneous" magnetization in the absence of a magnetic field. Thus, we consider the revised sentence properly describe the current situation of multiferroic research field.

In addition, we replaced the description "still quite rare" by "not common" in the third sentence of the introduction section (p. 2).

We also italicize the word "same" in the first sentence of the last paragraph in p. 2 in order to emphasize the rare condition with $P||M$ in spin-driven ferroelectrics.

(1-2) *The detailed description about the M_Q^{D1} and M_Q^{D1} ($Q=Q_A$ or Q_B) order parameters should be provided. How to get the transformations (in supplementary materials) of these four parameters also need to be explained in detail. This will be helpful to various reader to*

understand their phenomenological model. By the way, will it be possible to show a schematic illustration about such transformations?

In this paper, M_Q^{D1} and M_Q^{D2} represent the amplitude of the magnetization distribution which transforms according to the irreducible corepresentations D1 and D2 for the space group $Pnma$ and $Q_{ic} = (q_h q_k 0)$ [ref. 18 in the revised manuscript]. For example, the distribution associated with the order parameter M_Q^{D1} is even under m_z , whereas that associated with M_Q^{D2} changes sign under m_z . Results for other operators are given in equations (S4)-(S7) of Supplementary Table 1.

Following the reviewer's suggestion, we inserted the following sentence at the third sentence of "Phenomenological coupling theory" section (p. 4).

" M_Q^{D1} and M_Q^{D2} represent the amplitude of the magnetization distribution which transforms according to the irreducible corepresentations D1 and D2 for the space group $Pnma$ and $Q_{ic} = (q_h q_k 0)$ (ref. 18), e.g. M_Q^{D1} is even under m_z and M_Q^{D2} is odd under m_z ."

This comment is related to the comment (3-1) raised by Reviewer 3. Please see also our reply to the comment (3-1).

We considered the reviewer's suggestion about a schematic illustration. However, the spin functions we are dealing with are very complicated, and we think that the algebraic exposition is the most definitive. To state that taking z into $-z$ takes $f(x,y,z)$ into $f(x,y,-z)$ is easier for the reader than diagrams to illustrate this. Therefore, we prefer not to have to try to implement this idea.

(1-3) In Fig. 1c, I suggest the authors to mark the directions of polarization and magnetization, respectively. If possible, the (magnetic) space group or point group resulted by the corresponding magnetic configurations can also be added in Fig. 1c.

The magnetic structure of Mn_2GeO_4 is composed of both incommensurate and commensurate orders, and therefore is very complicated. Due to the magnetic structure, only the operation $2_1 \parallel c$ can remain as a symmetry element, which means that the point group symmetry lowers from mmm to 2. Indeed, this satisfies the symmetry requirement for magnetic point groups of ferromagnetic and ferroelectric structure with $M \parallel z$ and $P \parallel z$ ("2" or "1" for the present case) [Cracknell, A. P. Magnetism in crystalline materials (Pergamon Press, Oxford 1975)].

To the best of our knowledge, incommensurate magnetic structures are not described by a magnetic space group, but with a magnetic superspace group in four dimensions. Therefore, we have not determined the magnetic space group.

Following the reviewer's suggestion, we marked the directions of polarization and magnetization in the revised Fig. 1c, and added a description about the magnetic point group "(the most probable magnetic point group: 2 with 2_1 axis along c)" in the figure caption of Fig. 1c.

(1-4) Will it be possible to provide a schematic illustration to show how the inversion center is broken by the magnetic configurations in Fig 1c?

In the *Pnma* crystal structure of Mn_2GeO_4 , the inversion center is located at Mn1 site. Thus, it is apparent that the magnetic configurations break the inversion symmetry. We consider that schematic illustrations of the inversion centers in Fig. 1c make the figure confused, and don't want to add such an illustration.

(1-5) In line 190, typo--"profiles of the $(-3\ 1\ 0)$, $(-3\ 1\ 0)$, $(-2+qh\ -1+qk\ 0)$, and $(-2+qh\ 1-qk\ 0)$ reflections"

Thank the reviewer for his/her critical reading and finding the typo. We corrected the typo in the revised manuscript.

(1-6) Minor suggestions:

(i) In Supplementary Table 1, I understand that considering the mirror symmetry operations (m_x , m_y and m_z) are enough to derive the transformations of *Pnma* symmetry, because the other operations I , 2_x , 2_y and 2_z can be generated by (m_x , m_y and m_z). I still suggest the authors to indicate why considering only (m_x , m_y and m_z) symmetry operations will be enough to derive the invariant terms;

(ii) In Supplementary Table 1, I understand that the polar vectors (e.g. P) and axial vectors (e.g. M) transform in the same way under rotations, but in different ways under mirrors. I still suggest the authors to write a little bit about such transformation rule.

Following the reviewer's suggestions, we added the following descriptions at the caption of Supplementary Table 1.

"The transformation properties of the order parameters, given below (S1-S7), were developed by considering how the magnetization distribution transforms under the symmetry operations of the space group, similar to the procedure used in ref. 29. The mirror operations, m_x , m_y , and m_z , are the minimum symmetry elements for point group *Pnma*, because all the other operations (I , 2_x , 2_y , and 2_z) can be generated by the product of these mirror operations. Note that the transformations of axial vectors such as the magnetization, magnetic order parameters, or the angular momentum $\mathbf{r} \times \mathbf{p}$ are different from those of polar vectors such as position \mathbf{r} and momentum \mathbf{p} . For axial vectors, the result for transformations involving a change in handedness (e. g. mirror or inversion operation) includes an extra factor of (-1) ."

[Response to Reviewer #2]

(2-1) I have to still mention a problem in the author's view reflected in the manuscript though. As the MF research gets more and more ample, we all know that the understanding

of the mechanism in each rare case of ferromagnetic ferroelectric MF becomes increasingly important. The authors seems to address well this issue in the abstract and introduction while they explain the cases of the hexaferrites and CoCr₂O₄. However, the authors seem to describe only the author's works without unbalanced citations of the other progresses in the hexaferrites, particulaly the ones with Al doped compounds. I hope this can be remedied in their manuscript before being published.

In the previous manuscript, we cited ref. 10 (Kimura, Annu. Rev. Condens. Matter Phys.) written by one of us as a REVIEW paper on magnetoelectric hexaferrites relevant to their conical spiral magnetic structures. This review cited various literatures related to this topic. However, the referee considers that such citation is unbalanced. Following the referee's suggestion, we added the following original papers dealing with hexaferrites showing the magnetoelectric effect relevant to conical-spiral magnetic structures.

11. Ishiwata, S., Taguchi, Y., Murakawa, H., Onose, Y. & Tokura, Y. Low-magnetic-field control of electric polarization vector in a helimagnet. Science 319, 1643-1646 (2008).

13. Chun, S. H. et al. Realization of giant magnetoelectricity in helimagnets. Phys. Rev. Lett. 104, 037204 (2010).

14. Tokunaga, Y. et al. Multiferroic M-type hexaferrite with a room-temperature conical state and magnetically controllable spin helicity. Phys. Rev. Lett. 105, 257201 (2010).

15. Soda, M., Ishikura, T., Nakamura, H., Wakabayashi, Y. & Kimura, T. Magnetic ordering in relation to the room-temperature magnetoelectric effect of Sr₃Co₂Fe₂₄O₄₁. Phys. Rev. Lett. 105, 257201 (2010).

16. Wang, F., Zou, T., Yan, L.-Q., Liu, Y. & Sun, Y. Low magnetic field reversal of electric polarization in a Y-type hexaferrite. Appl. Phys. Lett. 100, 122901 (2012).

[Response to Reviewer #3]

(3-1) *Section Phenomenological coupling theory: The authors write “We construct coupling terms that the reader can check are invariant under these transformation properties, and this leads to three such terms given by...”. While I have no doubt that the terms are indeed invariant under these transformation properties, I don't think it is the reader's responsibility to verify this. I strongly suggest that the authors include a section in the Supplementary Material that demonstrates that this is the case.*

As pointed out by the reviewer, it may be better for readers' understanding to describe some more details how the phenomenological coupling theory was developed. In this reply, we

attach, for the reviewer's eyes only, a preprint written by one of the authors (Harris_preprint.pdf). This preprint provides complete details on the derivation of the transformation properties of the order parameter and will be submitted to as soon as the present paper is available to the public. From the preprint, the reviewer might be able to judge the reliability of our phenomenology. However, we consider that it is impractical to give the full details of the derivation in the present paper. Instead, we inserted the following subsection in Methods section (p. 11).

“Phenomenological coupling theory. In order to develop the phenomenological coupling theory, we first determined the symmetry properties of the magnetic and ferroelectric order parameters found experimentally. The symmetry properties of the magnetic order parameters are determined by their irreducible representation, and leads directly to the transformation properties of the order parameters given in equations (S2)-(S7) of Supplementary Table 1, similar to the procedure used in ref. 29. The transformation property of the ferroelectric polarization is given in equation (S1) of Supplementary Table 1. Using these transformation properties, we construct coupling terms that are invariant under the space group symmetry elements, and three of those are listed in equations (1)-(3). A full account of the phenomenological coupling theory will be published elsewhere.”

In addition, we revised the sentence just above equation (1) on p. 4, as follows.

“We construct coupling terms that the reader can check are invariant under these transformation properties by using equations (S1)-(S7) in Supplementary Table 1,…”

(3-2) Section Electric-field-cooling effect on spin-helicity: To the expert reader, familiar with the SNP technique, it is immediately clear that measuring the (z, x) and $(z, -x)$ polarization channels provides a direct way of determining the spin helicity. However, to the broader audience, and even to most researchers familiar with neutron scattering this is not clear. After mentioning that these polarized channels were determined for varying the strength of the electrical field applied during cooling, the authors go on to describe further technical details, and only on the next page, explain that measuring these cross-sections, and eventually the full polarization matrix allows to determine the population of the two spin-helicity domains. To highlight the importance of this measurement to the non-expert reader and to improve the flow of the text, I recommend adding a statement at the beginning of the paragraph that explains that measuring the (z, x) and $(z, -x)$ polarization channels is a direct way of determining spin helicity and is, thus, the method of choice to understand the nature of the U coupling term. I note that, for example, in the section Correlation between the C and Q domains the authors have done a much better job of explaining at the beginning of the section what kind of information can be obtained via the complimentary unpolarized neutron diffraction measurements.

Please see our reply to the comment (3-3).

(3-3) In Fig. 2a, as well as the corresponding discussion in the text, the polarized scans of the magnetic Bragg peaks are called “*k*-scan profiles”. I assume that the authors want to imply that they are scanning over the peaks keeping the *h* and *l* components for the momentum transfer constant while varying *k* component. I think that for someone experienced in neutron diffraction this is clear, however, for a non-expert “*k*-scan profiles” is a confusing name, notably, because they will not be familiar with the fact that in scattering techniques the three components of the momentum transfer in reciprocal lattice units are typically called (*h*, *k*, *l*). I would very much prefer to call the scans *Q*-scans (because in the method section the momentum transfer is called *Q*) or even better momentum transfer scans along the *k* direction. I would further replace the label in the upper left corner of Fig. 2a to be (2-*q_h* *k* 0).

Following the reviewer’s suggestions, we revised the following parts.

The third sentence of the section “Electric-field-cooling effect on spin-helicity” (p. 5) was revised as follows.

“The intensity profiles of the ($\pm 2\mp q_h$ $1-q_k$ 0) peaks were measured in both the non-spin-flip (*z,x*) and spin-flip (*z,-x*) polarization channels. From the difference in the intensities observed in each polarization channel, the relative proportions of the two helicity domains within each *Q* domain can be determined directly (see Methods).”

In the fifth sentence of the section “Electric-field-cooling effect on spin-helicity” (p. 5), we replaced the term “*k*-scan profiles obtained” with “the wave-vector dependence of the scattering”.

In the figure caption of Fig. 2a, we replaced the description “Polarized neutron scattering measurements of the *k*-scan profiles” with “Polarized neutron scattering as a function of wave-vector transfer along the (0,*k*,0) wave-vector direction”.

(3-4) The authors highlight twice in the manuscript that no clear microscopic explanation for the simultaneous reversal of the magnetization and polarization observed in Mn₂GeO₄ and several other multiferroic materials exists to date. The discussion is written in a way that suggests that the authors established a microscopic theory that explains such switching for Mn₂GeO₄. In contrast, the theory presented in the paper is only phenomenological and based on symmetry considerations. The “microscopic” explanation provided in the discussion of the paper is based on the magnetic structure, which consists of two coupled canted antiferromagnetic conical spin-chains. The superposition of those two chains leads to net polarization and magnetization along the *c* axis. The neutron diffraction results described in the results part of the text show that application of a magnetic field along *c* leads to a flop of the cone axis in each chain, and thus simultaneous reversal of both polarization and magnetization. While the authors observe the microscopic magnetic structure of Mn₂GeO₄ and its change as function of applied magnetic and electrical field, this does not provide a microscopic explanation based on a microscopic Hamiltonian. Basically, the microscopic interactions that lead to the observed magnetic structure and its behavior in applied fields is not provided by the authors. The authors say that the polarization in the individual chains is

well-explained within inverse Dzyaloshinskii-Moriya mechanism. This indeed makes sense in terms of their experimental observations. However, in that sense there is no new insights on how this simultaneous reversal of polarization and magnetization works, as this has been already proposed earlier for other materials. What is distinct about Mn₂GeO₄ seems to be rooted in details of its magnetic structure, namely that it is composed in a special superposition of two canted antiferromagnetic conical spin-chains. I thus feel that the manuscript needs some fine-tuning that is less suggestive of the authors identifying a new level of microscopic understanding of the simultaneous reversal of the magnetization and polarization in multiferroic materials.

The reviewer considers that some descriptions about the microscopic understanding of the observed M-P coupling is a bit too strong. We basically agree to the reviewer's comment on this point. Therefore, following the reviewer's suggestion, we fine-tuned some descriptions.

At the end of the first paragraph of introduction (p. 2), we added the following sentence.

“An important result of the present paper is to show that the phenomenological model we introduce here explains the switching mechanisms we observe in the double-Q conical-spiral multiferroic, Mn₂GeO₄.”

At the third last sentence of the first paragraph on p. 3 (“Although phenomenological symmetry...”), the term “no clear microscopic mechanism” was replaced by “no clear explanation”.

The word “microscopic” at the last sentence of the second paragraph on p. 9 (“The term W describes...”) was removed.

The word “microscopically” at the last sentence of the summary paragraph in the previous manuscript (“Furthermore, we revealed...”) was removed.

By these revisions, the manuscript becomes less suggestive of our identifying a new level of microscopic understanding of the simultaneous reversal of the magnetization and polarization in multiferroic materials.

(3-5) For the summary paragraph, I wonder if there are some “lessons” or “guidelines” that can be extracted from the more detailed understanding of the coupling of the magnetization and electrical polarization in Mn₂GeO₄. For example, is it expected that such coupling based on coupled spin chains is found in more materials? Or is this a specific case?

Responding to the reviewer's comment, we added the following statement at the end of summary paragraph (p. 10).

The present study conclusively describes how magnetization and ferroelectric polarization can be coupled on a phenomenological level and stands out as a text-book case of how complex and seemingly unexpected coupling terms can be engineered in spin-driven ferroelectrics with double-Q conical magnetic order. The essence of the present phenomenological discussion can be also applied to other conical-spin-driven ferroelectrics.

REVIEWERS' COMMENTS:

Reviewer #1 (Remarks to the Author):

My comments raised in the previous round have been satisfactorily addressed by the authors. I understand the authors' situation that the schematic illustrations (about the breaking of inversion center and order parameter transformations) will make the figures confused, because the incommensurate magnetic structures are complicated. Thus, I now recommend this manuscript for publication.

Reviewer #2 (Remarks to the Author):

The modified manuscript improves some of vague and omitted explanations read in the original version. By its own nature of describing the complex coupling of parallel M and P, it is still uncomfortable to read this manuscript itself. This has been partly unavoidable because a separate theory paper is required to understand fully, regardless of the present one, which is rather a disappointing part, but still acceptable in my part.

Besides, I believe the present case presents one solution to understand how the parallel M and P coupling can be understood in spin-driven ferroelectrics, particularly one generated by conical spin order where usually perpendicular relationship of M and P is realized. Therefore, the last paragraph the authors have added reads as a too much stretch to the general case. Therefore, the sentence or paragraph starting with "The present case conclusively..." should contain some descriptions that this work specifically explains the case where M and P is parallel in materials with conical spin orders (rather than stating "double-Q" as a specific word of this case)..

Reviewer #3 (Remarks to the Author):

I have reviewed the changes the authors have made in response to all reviewer comments and I agree that the manuscript can now be published in Nature Communications as is.

[Response to Reviewer #2]

We would like to sincerely thank the referee for a careful reading and valuable comments on our manuscript.

(Comment) Besides, I believe the present case present one solution to understand how the parallel M and P coupling can be understood in spin-driven ferroelectrics, particularly one generated by conical spin order where usually perpendicular relationship of M and P is realized. Therefore, the last paragraph the authors have added read as a too much stretch to the general case. Therefore, the sentence or paragraph starting with "The present case conclusively..." should contain some descriptions that this work specifically explains the case where M and P is parallel in materials with conical spin orders (rather than stating "double- Q " as a specific word of this case).

Following the referee's suggestion, we revised the paragraph as follows.

“The present study describes how magnetization and ferroelectric polarization in spin-driven ferroelectrics with conical magnetic order can be coupled on a phenomenological level in the case where they are parallel to each other. Furthermore, it stands out as a text-book case of how complex and seemingly unexpected coupling terms can be engineered in such spin-driven ferroelectrics. The essence of the present phenomenological discussion can be also applied to other conical-spin-driven ferroelectrics.”